# AC-Foley: Reference-Audio-Guided Video-to-Audio Synthesis with Acoustic Transfer

Pengjun Fang[1], Yingqing He[1], Yazhou Xing[1], Qifeng Chen[1,✉], Ser-Nam Lim[2,✉], and Harry Yang[1,✉]

[1]The Hong Kong University of Science and Technology
[2]University of Central Florida

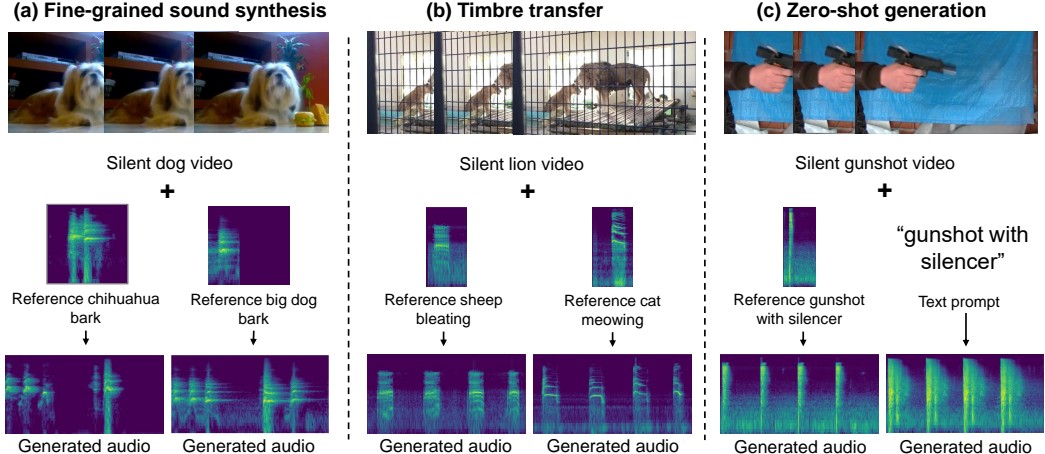

Figure 1: **AC-Foley for conditional Foley generation with audio controls.** (a) Fine-grained sound synthesis: AC-Foley generates precise audio from a silent dog video based on reference sounds, such as a Chihuahua's or a big dog's bark. (b) Timbre transfer: Given a silent lion video, AC-Foley produces different audio outputs conditioned on reference sounds, such as sheep bleating or a cat meowing. (c) Zero-shot generation: Given a silent gunshot video, AC-Foley generates a gunshot with a silencer with reference audio, while a text prompt fails to do so.

## Abstract

Existing video-to-audio (V2A) generation methods predominantly rely on text prompts alongside visual information to synthesize audio. However, two critical bottlenecks persist: semantic granularity gaps in training data (e.g., conflating acoustically distinct sounds like different dog barks under coarse labels), and textual ambiguity in describing microacoustic features (e.g., "metallic clang" failing to distinguish impact transients and resonance decay). These bottlenecks make it difficult to perform fine-grained sound synthesis using text-controlled modes. To address these limitations, we propose **AC-Foley**, an audio-conditioned V2A model that directly leverages reference audio to achieve precise and fine-grained control over generated sounds. This approach enables: fine-grained sound synthesis (e.g., footsteps with distinct timbres on wood, marble, or gravel), timbre transfer (e.g., transforming a violin's melody into the bright, piercing tone of a suona), zero-shot generation of sounds (e.g., creating unique weapon sound effects without training on firearm datasets) and better audio quality. By directly conditioning on audio signals, our approach bypasses the semantic ambiguities of text descriptions while enabling precise manipulation of acoustic attributes. Empirically, AC-Foley achieves state-of-the-art performance for Foley generation when conditioned on reference audio, while remaining competitive with state-of-the-art video-to-audio methods even without audio conditioning.

# 1 INTRODUCTION

Current video-to-audio generation frameworks aim to synthesize sound effects that are temporally and semantically aligned with the video to perform Foley tasks (Wang et al., 2024a; Cheng et al., 2024; Liu et al., 2024; Viertola et al., 2025; Wang et al., 2024b; Zhang et al., 2024). While these approaches have made progress in generating synchronized audios, they often fail to provide the fine-grained control needed by sound creators. They cannot synthesize creator-specified variations – a limitation starkly evident when artists need multiple acoustic versions of the same visual action (e.g., footsteps varying by surface material). Most existing systems provide only limited control mechanisms, including video clip conditions (Du et al., 2023) and text (Xie et al., 2024), but these approaches face two fundamental limitations: 1) Dataset granularity gaps: Training annotations often flatten acoustically distinct categories (e.g., labeling all dog vocalizations as "barking"). Consequently, even with differentiated prompts like "high-pitched Chihuahua bark" versus "deep German Shepherd growl", models generate sonically indistinguishable outputs due to insufficient acoustic diversity in supervision. 2) Descriptive limitations of language: Text prompts inherently fail to encode micro-acoustic attributes – for instance, "metallic clang" ambiguously represents both a hammer striking an anvil (sharp attack, high-frequency resonance) and a steel chain dropping (diffused impact, low-mid decay), resulting in inconsistent audio rendering. These constraints severely restrict the ability to specify nuanced sound variations aligned with creative intent.

To address these limitations, some recent works have attempted to improve flexibility by enhancing text control for audio generation or doing audio extension based on audio conditions (Chen et al., 2024). However, text-based methods remain constrained by language's inability to specify sub-semantic acoustic details, while audio extension approaches inherently limit creative diversity by anchoring outputs to pre-existing sounds. This leaves creators without tools to synthesize novel yet precisely controlled audio aligned with artistic vision.

In this work, we propose a reference-audio guided video-to-audio synthesis framework to bridge this gap. By integrating reference audio as a control signal, our method enables precise sound characteristic manipulation while maintaining synchronization, avoiding semantic ambiguity in text through direct acoustic modeling. Building on multimodal joint training following (Cheng et al., 2024), we unify video, audio, and text modalities to learn cross-modal representations that enhance both quality and controllability. Empirically, we observe a significant relative improvement in audio quality (20% lower Fréchet Distance (Kilgour et al., 2019) and 28% lower Kullback–Leibler distance) and acoustic fidelity (22% lower Mel Cepstral Distortion).

Previous work (Du et al., 2023) shares some similarities with ours by also incorporating audio as a control mechanism. However, their method requires a reference video clip (including audio) for control, and the reference and generated audio must have identical durations, limiting flexibility. Additionally, their approach was trained on relatively small datasets (Greatest Hits (Owens et al., 2016) and Countix-AV (Zhang et al., 2021)), which restricts generalizability compared to our framework.

The central challenge of our method is adapting reference audio to the video context without sacrificing synchronization or audio quality. Simply overlaying the reference sound onto the footage leads to two main problems: temporal misalignment (mismatched duration and pacing) and poor audio–visual cohesion when the sound is not properly adapted. This is especially difficult when the system must both generate sounds that are synchronized with visual events and transform the conditional reference audio to match the video's timing while preserving its timbral characteristics. In short, the difficulty lies in learning how to transform the reference audio to fit the temporal and contextual structure of video, ensuring that the resulting audio is both coherent with the visuals and faithful to the characteristics of the reference sound. This underscores the need for innovative methods capable of bridging this gap.

Our solution introduces a two-stage training framework: 1) Acoustic Feature Learning: Train with overlapping audio-video segments to establish reference sound feature extraction. 2) Temporal Adaptation: Condition on non-overlapping audio from the same video, leveraging inherent audio self-similarity (e.g., footsteps in a scene share acoustic properties). This phase forces the model to align reference characteristics with visual timing while preserving acoustic fidelity.

In summary, we propose **AC-Foley**, a video-to-audio synthesis framework enabling precise acoustic control via reference audio conditioning. By unifying video, audio, and text modalities through joint

training, our method learns adaptive cross-modal representations that preserve synchronization while transforming reference sounds to match video context.

## 2 RELATED WORK

### 2.1 VIDEO-TO-AUDIO GENERATION

Recent progress in multimodal generation has spurred diverse technical approaches for video-conditioned audio synthesis. Transformer-based architectures dominate the field, with methods like SpecVQGAN (Iashin & Rahtu, 2021), FoleyGen (Mei et al., 2024b) and V-AURA (Viertola et al., 2025) employing auto-regressive frameworks for temporal coherence, while some methods (Liu et al., 2024; Pascual et al., 2024; Tian et al., 2025) utilize masked token prediction for audio waveform generation. An emerging paradigm leverages diffusion models and flow matching techniques, such as the latent space denoising mechanisms of Diff-Foley (Luo et al., 2023) and VTA-LDM (Xu et al., 2024) and the rectified flow matching of Frieren (Wang et al., 2024b). Some approaches (Jeong et al., 2025; Wang et al., 2024a; Xing et al., 2024; Zhang et al., 2024) train new control modules for pre-trained text-to-audio models on audio-visual data to perform video-to-audio tasks, and recent works like Movie Gen Audio (Polyak et al.) demonstrate text's complementary role in video-conditioned synthesis. Though these methods achieve varying degrees of synchronization, they primarily focus on reproducing audio semantically implied by visual content. MMAudio (Cheng et al., 2024) explores multimodal joint training across video and text modalities but remains limited to basic semantic control. Our approach advances this field by enabling precise acoustic manipulation through audio conditioning while maintaining synchronization, supporting novel Foley applications like semantic sound substitution and timbre transfer that existing methods cannot achieve.

### 2.2 TIMBRE CONTROL

Prior audio manipulation research primarily focused on single-modality transformations. Early style transfer methods adapted image synthesis techniques like feature statistic matching to separate audio content from timbral style (Verma & Smith, 2018). Musical timbre editing frameworks (Huang et al., 2018) leveraged CycleGAN (Zhu et al., 2017) architectures for cross-instrument sound conversion. While effective for audio-to-audio tasks, these methods ignore visual context crucial for video-synchronized Foley applications. Recent video-aware approaches introduce novel conditioning paradigms: MultiFoley (Chen et al., 2024) extends partial audio tracks into complete soundscapes while preserving original acoustic signatures through audio continuation, and CondFoley (Du et al., 2023) generates analogous sounds by matching full-length audio-video pairs. However, fundamental limitations persist – audio extension methods constrain output diversity through strict inheritance of conditioned clips, while duration-matched conditioning restricts creative adaptation across temporal scales. Our approach transcends these constraints by enabling variable-length audio conditioning without temporal coincidence requirements, achieving both precise timbral control and flexible synchronization with visual events.

## 3 AC-FOLEY

### 3.1 PRELIMINARIES

**Conditional Flow Matching Objective.** We extend conditional flow matching (CFM) (Lipman et al., 2022; Tong et al., 2023) to jointly model three modalities: video $\mathbf{V}$, audio $\mathbf{A}$, and text $\mathbf{T}$. The enhanced velocity field $v_\theta$ now operates under the multimodal condition $\mathcal{C} = \{\mathbf{V}, \mathbf{A}, \mathbf{T}\}$ through

$$\mathbb{E}_{t,q(x_0),q(x_1,\mathcal{C})}\|v_\theta(t,\mathcal{C},x_t) - (x_1 - x_0)\|^2, \tag{1}$$

where timestep $t \in [0,1]$, $q(x_0)$ is the standard normal distribution, $q(x_1,\mathcal{C})$ is sampled from training data, and $x_t = tx_1 + (1-t)x_0$ linearly interpolates between Gaussian noise $x_0$ and target latent $x_1$.

### 3.2 MULTIMODAL TRANSFORMER.

Our objective is to synthesize temporally precise and acoustically faithful sound effects for silent videos through multimodal conditional guidance. Formally, given a silent video sequence $\mathbf{V} \in$

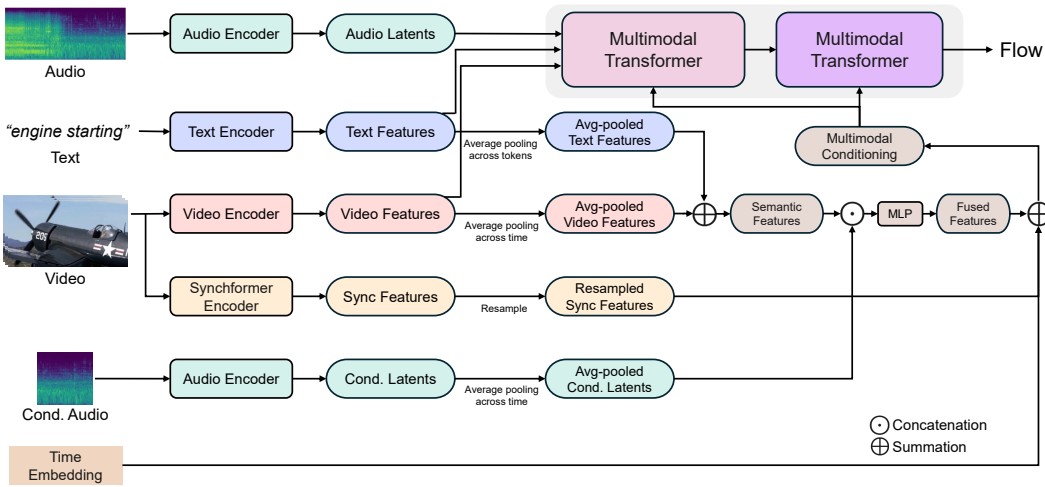

Figure 2: **Overview of our method.** Different modalities (video, text, and audio) jointly interact in the multimodal transformer network. Multimodal conditioning with audio injects semantic, temporal and acoustic information for more precise control.

$\mathbb{R}^{T_v \times H \times W \times 3}$ with $T_v$ frames, a reference audio clip $\mathbf{A}_c \in \mathbb{R}^{T_a}$ specifying target acoustic properties and a text prompt $\mathbf{T}$ describing semantic requirements, we learn a conditional generation model $\mathcal{G}_\theta$ that produces

$$\mathbf{A}_t = \mathcal{G}_\theta(\mathbf{V}, \mathbf{A}_c, \mathbf{T}) \quad \text{where} \quad \mathbf{A}_t \in \mathbb{R}^{T_a}. \tag{2}$$

As illustrated in Figure 2, we adopt the successful framework of the multimodal transformer design, which can efficiently model the interactions between video, audio, and text modalities.

### 3.3 AUDIO CONTROL MODULE

**Audio Encoding.** The audio processing pipeline begins by converting raw waveform signals into time-frequency representations through Short-Time Fourier Transform (STFT) operations. Following this, we compute mel-scale spectral (Stevens et al., 1937) representations that serve as intermediate features. These spectral features undergo dimensional reduction via a pretrained variational autoencoder (VAE) (Kingma & Welling, 2014), producing compact latent embeddings $x_1$ that drive our generation process.

During the synthesis phase, the system reconstructs audio outputs through a two-stage inversion process: First, the generated latent vectors are projected back to mel-spectrogram space using the VAE decoder. Subsequently, these reconstructed spectral representations are converted into time-domain waveforms through a pretrained vocoder (Lee et al., 2022).

**Multimodal Conditioning with Audio.** Our conditioning mechanism addresses the limitations of existing methods, which primarily rely on text or video for control. While some approaches (Lee et al., 2025) incorporate conditional audio inputs, they often use encoders like CLAP (Wu et al., 2023) to process the audio, extracting only semantic information and overlooking the rich acoustic features present in the audio signal. We use the pretrained VAE encoder for processing reference audio, which preserves the complete acoustic signature (spectral/timbral characteristics) through its latent space.

In our method, we compute a multimodal conditioning vector $\mathbf{c} \in \mathbb{R}^{1 \times h}$ shared across all transformer blocks, which integrates information from text, video, and conditional audio. The conditional audio is processed through our audio encoding pipeline, followed by average pooling, to extract meaningful acoustic features that capture fine-grained auditory details. These acoustic features are combined with the Fourier encoding of the flow time step, the visual and text features encoded by CLIP (Radford et al., 2021) and average-pooled, and the sync features (initially extracted at 24 fps by Synchformer (Iashin et al., 2024) and resampled via nearest-neighbor interpolation to match the audio latent representation) to form the multimodal conditioning vector $\mathbf{c}$ (Figure 2).

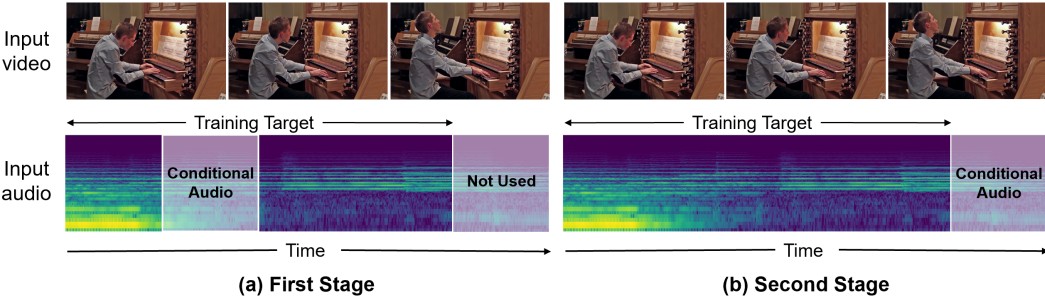

Figure 3: **Illustration of the two-stage training process for audio generation.** (a) Stage I: Overlapping Conditioning. The random 2 seconds of the 8-second target audio are used as the conditional audio, allowing the model to learn the utilization of acoustic features from overlapping audio segments. (b) Stage II: Non-overlapping Conditioning. The non-overlapping last 2 seconds of the 10-second video clip are used as the conditional audio, leveraging inherent audio self-similarity within the video to enhance model generalization.

This multimodal conditioning vector is then applied to modulate the input $\mathbf{f} \in \mathbb{R}^{L \times h}$, where $L$ is the sequence length, using adaptive layer normalization (adaLN) layers (Perez et al., 2018):

$$\text{adaLN}(f, c) = \text{LayerNorm}(f) \cdot \mathbf{W}_\gamma(c) + \mathbf{W}_\beta(c), \tag{3}$$

where $\mathbf{W}_\gamma$ and $\mathbf{W}_\beta$ are MLPs. By explicitly incorporating acoustic features from the conditional audio, rather than relying solely on semantic information, our method provides richer and more precise control over audio generation. This design enables the model to leverage both the semantic context and the detailed acoustic characteristics of the input, resulting in more contextually and acoustically aligned outputs.

## 3.4 TRAINING STRATEGY

Following MMAudio (Cheng et al., 2024), we train our model on both audio-text-visual datasets and audio-text datasets. Specifically, we use VGGSound (Chen et al., 2020), which contains approximately 180K 10-second videos, as our audio-text-visual dataset. For audio-text datasets, we utilize AudioCaps2.0 (Kim et al.), comprising around 98K manually captioned 10-second audio clips, and WavCaps (Mei et al., 2024a), which includes roughly 7600 hours of automatically captioned audio. Since the audio clips in WavCaps vary in length, we extract non-overlapping 10-second segments, resulting in a combined total of 600K audio-text pairs, including data from AudioCaps2.0.

**Two-Stage Training.** We adopt a two-stage training scheme. From each 10-second video clip, we take the first 8 seconds as the training target. In Stage I (overlap), we randomly sample a 2-second segment from those 8 seconds to serve as the conditional audio (Figure3a). This direct reference–target alignment teaches the model to extract and exploit acoustic features (e.g., timbre and spectral patterns), but because the condition overlaps the target, it can encourage trivial "copy and paste" behavior. To mitigate that, in Stage II (no overlap), we use the last 2 seconds of the 10-second clip, which does not overlap the 8-second target, as the condition (Figure3b). This exploits the natural self-similarity often present within videos (e.g., repeated actions) and forces the model to apply learned acoustic features in novel temporal contexts rather than simply reproducing the reference.

This complementary design addresses the main failure modes of single-stage approaches: overlap-only training yields reference-replicating behavior, while non-overlap-only training creates a feature-utilization gap and temporal disconnection because aligned reference–target pairs are absent. Stage I supplies synchronized supervision for reliable feature extraction; Stage II enforces generalization and prevents reliance on overlap.

Finally, we finetune our model for 40k iterations on a high audio-visual correspondence subset of VGGSound (Chen et al., 2020), which was selected using an ImageBind (Girdhar et al., 2023) score threshold of 0.3, following (Viertola et al., 2025; Chen et al., 2024).

Through this two-stage training approach, we find that the model learns to assume that the conditional audio is informative about the target sound. Empirically, this leads the model to base its predictions on the conditional sound rather than on simple overlap. As a result, at test time, the model can generate high-quality audio even when the conditional sound is sampled from a completely different video.

## 4 EXPERIMENTS

### 4.1 EXPERIMENT SETUP

We assess our model using the VGGSound test set (Chen et al., 2020), refining the dataset by employing ImageBind (Girdhar et al., 2023) to exclude samples with a correspondence score below 0.3, following (Viertola et al., 2025; Chen et al., 2024). This process results in a curated set of 8,676 videos. For each 10-second video, we extract the first 8 seconds of the video as video input and use the final 2 seconds of the original audio as conditioning input. Notably, using the final 2s as a non-overlapping reference does not introduce bias, since 10s clips are typically trimmed from longer continuous videos/audios, which means the last 2s are not systematically different from other segments. For fair evaluation, all audio generations are assessed at the 8-second mark. We compare our model against various video-to-audio synthesis baselines, utilizing precomputed samples from MultiFoley (Chen et al., 2024), Frieren (Wang et al., 2024b), and reproducing results using the official inference code for MMAudio (Cheng et al., 2024), FoleyCrafter (Zhang et al., 2024), V-AURA (Viertola et al., 2025), SSV2A (Guo et al., 2024), ThinkSound (Liu et al., 2025) and HunyuanVideo-Foley (Shan et al., 2025).

### 4.2 METRICS

Following prior works (Cheng et al., 2024; Chen et al., 2024), we evaluated our model's performance across several dimensions: distribution matching, semantic alignment, temporal synchronization, and spectral fidelity—the latter to account for the control of acoustic characteristics through conditional audio. We employed Fréchet Distance (FD) and Kullback–Leibler (KL) distance to assess distribution matching, utilizing PaSST (Koutini et al., 2021), PANNs (Kong et al., 2020), and VGGish (Gemmeke et al., 2017) as embedding models for FD, and PANNs and PaSST as classifiers for the KL distance.

Semantic alignment was evaluated using the ImageBind (Girdhar et al., 2023) score, which measures the semantic correspondence between the generated audio and the input video. Temporal synchronization was evaluated using a synchronization score (DeSync), predicted by Synchformer (Iashin et al., 2024), which quantifies the misalignment (in seconds) between audio and video. Due to Synchformer's context window limitation of 4.8 seconds, we averaged the results from the first and last 4.8 seconds of each 8-second video-audio pair. As a complementary measure of temporal alignment, we also report onset accuracy, which is the proportion of correctly aligned audio event onsets between the generated and ground-truth audio, and its average precision (AP).

For spectral fidelity, we utilized Mel Cepstral Distortion (MCD) as our metric. A lower MCD value indicates a closer match between the synthesized and real mel cepstral sequences, suggesting higher fidelity in audio generation.

### 4.3 MAIN RESULTS

**Foley generation with audio conditioning.** Only one prior video-conditioned baseline (Video-Foley (Lee et al., 2025)) was available, but its performance was far from competitive. To create a stronger and fair comparison, we therefore train our own audio-conditioned baseline: we implement the MMAudio (Cheng et al., 2024) architecture and use CLAP (Wu et al., 2023) as the conditional audio encoder, keeping the same injection scheme and all training hyperparameters as our method. Under this controlled setup, AC-Foley outperforms both the trained MMAudio+CLAP baseline and the published Video-Foley model on all evaluation metrics, demonstrating that conditioning directly on acoustic features (our approach) offers advantages over using a semantic encoder like CLAP.

Compared to video-to-audio approaches more broadly, our method shows comprehensive advantages across distributional, semantic and spectral measures. Notably, while MMAudio (Cheng et al., 2024)

Table 1: Quantitative comparison of video-to-audio generation methods across multiple metrics. Best results are **bolded**; second-best results are underlined.

| Method | Distribution matching | | | | | Semantic | Temporal | | | Spectral |
|---|---|---|---|---|---|---|---|---|---|---|
| | $FD_{PaSST}\downarrow$ | $FD_{PANNs}\downarrow$ | $FD_{VGG}\downarrow$ | $KL_{PaSST}\downarrow$ | $KL_{PANNs}\downarrow$ | IB↑ | DeSync↓ | Onset Acc.↑ | Onset AP↑ | MCD↓ |
| With Audio Conditioning | | | | | | | | | | |
| Video-Foley | 613.05 | 73.17 | 17.45 | 4.16 | 4.75 | 3.6 | 1.214 | 0.2146 | 0.3409 | 17.41 |
| MMAudio + Clap | 70.80 | 7.95 | 4.33 | 1.17 | 1.36 | 35.7 | **0.431** | 0.2511 | 0.5107 | 14.63 |
| AC-Foley (ours) | **56.00** | **4.93** | **1.08** | **0.84** | **0.95** | **37.1** | 0.465 | **0.2832** | **0.5317** | **11.37** |
| Without Audio Conditioning | | | | | | | | | | |
| V-AURA | 215.95 | 14.55 | 2.40 | 1.66 | 1.99 | 31.1 | 0.947 | 0.2188 | 0.4880 | 15.52 |
| SSV2A | 236.71 | 17.47 | 2.34 | 1.74 | 1.85 | 26.2 | 1.210 | 0.2116 | 0.3988 | 19.79 |
| FoleyCrafter | 139.50 | 17.48 | 2.74 | 1.93 | 1.96 | 28.4 | 1.230 | 0.2033 | **0.5312** | 16.04 |
| Frieren | 110.61 | 11.29 | **1.38** | 2.46 | 2.36 | 25.5 | 0.856 | 0.2239 | 0.4689 | 14.98 |
| MultiFoley | 133.94 | 12.85 | 2.37 | 1.56 | 1.66 | 27.0 | 0.825 | 0.2431 | 0.5173 | 15.18 |
| ThinkSound (w/o. CoT) | 112.70 | 9.51 | 1.39 | 1.42 | 1.57 | 27.9 | 0.501 | 0.2735 | 0.5189 | 14.35 |
| HunyuanVideo-Foley | 85.19 | 12.14 | 2.91 | 1.52 | 1.72 | 34.7 | 0.492 | 0.2671 | 0.5271 | 15.12 |
| MMAudio-L-V2 | 69.25 | 8.81 | 3.98 | **1.12** | 1.34 | **37.8** | **0.392** | **0.2816** | 0.5257 | **14.11** |
| AC-Foley (w/o. audio) | **64.90** | **8.59** | 3.87 | 1.17 | **1.34** | 36.6 | 0.410 | 0.2619 | 0.5095 | 14.59 |

Table 2: Quantitative comparison of timbre transfer with audio conditioning on the Greatest Hits dataset. **Note that CondFoley is trained on the Greatest Hits dataset, while AC-Foley is not.**

| Method | Onset Acc. ↑ | Onset AP ↑ | MCD ↓ |
|---|---|---|---|
| CondFoley | 0.3906 | 0.6611 | 4.18 |
| AC-Foley (ours) | **0.3948** | **0.6629** | **3.39** |

achieves better DeSync scores, our investigation of ground truth (GT) audio-video pairs uncovers a DeSync mismatch of 0.558s, which is higher than the results of MMAudio and ours. This finding may imply that: (1) MMAudio and we may over-optimize for the Synchformer metric. (2) The metric's 4.8-second context window inadequately captures long-term synchronization patterns.

These comprehensive improvements suggest that AC-Foley achieves better holistic audio generation quality while maintaining precise control over acoustic properties - a critical requirement for video-conditioned audio synthesis tasks. Our findings particularly highlight the importance of unified feature representation learning, as evidenced by the consistent performance gains across complementary evaluation dimensions.

**Foley generation without audio conditioning.** Our framework can also support normal video-to-audio synthesis without audio condition. To achieve this, we replace the conditional audio input with a learned null embedding. We provide the results of our method comparison with the prior arts in Table 1. As shown in the table, our AC-Foley (w/o audio) achieves top or near-top performance on several distribution-matching metrics (lowest $FD_{PaSST}$ and $FD_{PANNs}$, tied/best $KL_{PANNs}$, and second-best $KL_{PaSST}$), while maintaining strong semantic alignment (IB second only to MMAudio-L-V2 (Cheng et al., 2024)) and temporal synchronization (DeSync near the best). Despite our primary focus being audio-conditioned generation, the unconditional (null-embedding) setting demonstrates that our framework can match or closely approach existing SOTA performance in video-to-audio tasks without fine-tuning.

**Timbre transfer with audio conditioning.** We evaluate our audio conditioning framework following the experimental protocol and dataset from (Du et al., 2023). The evaluation set is constructed from the Greatest Hits dataset (Owens et al., 2016), where 2-second silent video clips are randomly paired with three distinct 2-second conditional audio-visual clips from other test videos. We use onset accuracy, and its average precision (AP) to evaluate temporal synchronization. Mel-Cepstral Distortion (MCD) is used to measure acoustic fidelity.

As shown in Table 2, our AC-Foley outperforms CondFoley (Du et al., 2023) on all metrics, despite not being trained on the Greatest Hits dataset (Owens et al., 2016), unlike CondFoley. Additionally,

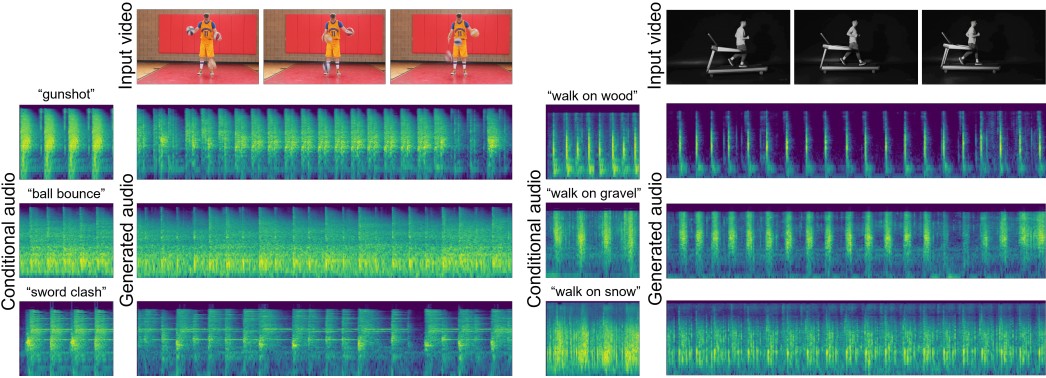

Figure 4: **Qualitative examples of Foley generation with audio conditioning**. We present generated results for two videos, each paired with three distinct conditional audio inputs. These examples highlight our model's ability to generate synchronized audio while adapting to varying acoustic characteristics, effectively demonstrating the impact of audio control.

Table 3: Comparison of our method and MMAudio-L-V2 in terms of temporal alignment and acoustic fidelity. We show our win rate and the tie rate of temporal alignment, and our win rate of acoustic fidelity. 95% confidence intervals are reported in gray.

| Comparison | Temporal alignment | | Acoustic fidelity |
|---|---|---|---|
| | Win rate (%) | Tie rate (%) | Win rate (%) |
| Ours **vs** MMAudio-L-V2 | 61.1$_{(\pm4.3)}$ | 21.8$_{(\pm3.6)}$ | 83.5$_{(\pm3.4)}$ |

while CondFoley requires conditional audio-visual clips to strictly match the duration of the generated audio, our framework supports flexible conditioning with arbitrary-length audio.

For fair comparison, we generate 2-second audio during testing, though our model is trained to handle 8-second sequences. This domain gap could slightly constrain our performance, yet we still achieve superior results. These improvements, combined with our flexible conditioning, highlight AC-Foley's robustness and generalization for real-world scenarios with variable condition lengths and limited domain-specific training data.

We also show some qualitative examples for Foley generation with audio conditioning in Figure 4, showcasing our model's ability to leverage the acoustic information from the conditional audio while maintaining precise temporal alignment. Please see our supplementary material for examples.

**Human studies.** We selected 32 high-quality videos from the VGGSound test set (Chen et al., 2020) to ensure a diverse range of categories and clear temporal information. For each video, we used the last 2 seconds of audio from the original 10-second clip as the conditional audio, with the corresponding category name serving as the text prompt to generate the audio for the first 8 seconds of the original video. Our method was compared against MMAudio-L-V2 (Cheng et al., 2024).

In the user study, participants watched and listened to three video clips for each question: one real clip and two generated clips. Each clip was paired with an audio sample—one corresponding to the real audio, one generated by our model, and the other produced by the baseline. Participants were asked to evaluate the following two aspects: (1) Acoustic Fidelity: Participants were instructed to select which generated audio was closer to the real audio. (2) Temporal Alignment: Given that both methods achieved good synchronization between audio and video, participants might find it challenging to determine which performed better. Therefore, in addition to the two options, we included the choice "Both have good sync / Difficult to choose." The results are presented in Table 3. For acoustic fidelity, our method significantly outperformed MMAudio-L-V2 (Cheng et al., 2024), achieving a win rate of 83.5%. In terms of temporal alignment, as both methods demonstrated similar performance, participants frequently selected the "Both have good sync / Difficult to choose" option

Table 4: Performance comparison of audio conditioning approaches (overlapping/non-overlapping segments) and finetuning strategies across distribution matching (FD/KL), semantic consistency (IB), temporal alignment (DeSync), and spectral quality (MCD) metrics.

| Method | Distribution matching | | | | | Semantic | Temporal | | | Spectral |
|---|---|---|---|---|---|---|---|---|---|---|
| | $FD_{PaSST}\downarrow$ | $FD_{PANNs}\downarrow$ | $FD_{VGG}\downarrow$ | $KL_{PaSST}\downarrow$ | $KL_{PANNs}\downarrow$ | IB↑ | DeSync↓ | Onset Acc.↑ | Onset AP↑ | MCD↓ |
| Overlap | 80.07 | 7.81 | 1.12 | 0.88 | 1.03 | 35.5 | 0.506 | 0.2502 | 0.5204 | 12.84 |
| Non-overlap | 60.82 | 5.06 | 1.20 | 0.84 | 0.96 | 36.8 | 0.506 | 0.2540 | 0.5206 | **11.30** |
| Two-stage w/o ft. | 56.00 | 5.11 | 1.21 | 0.84 | 0.95 | 37.0 | 0.468 | 0.2599 | 0.5229 | 11.37 |
| Two-stage | **56.00** | **4.93** | **1.08** | **0.84** | **0.95** | **37.1** | **0.465** | **0.2832** | **0.5317** | 11.37 |

Table 5: Results when we use average pooling or attention-based pooling.

| Method | Distribution matching | | | | | Semantic | Temporal | | | Spectral |
|---|---|---|---|---|---|---|---|---|---|---|
| | $FD_{PaSST}\downarrow$ | $FD_{PANNs}\downarrow$ | $FD_{VGG}\downarrow$ | $KL_{PaSST}\downarrow$ | $KL_{PANNs}\downarrow$ | IB↑ | DeSync↓ | Onset Acc.↑ | Onset AP↑ | MCD↓ |
| Attention-Based | **55.60** | 5.16 | 1.24 | **0.82** | 0.95 | 37.0 | 0.484 | 0.2598 | 0.5155 | **11.36** |
| Average (ours) | 56.00 | **4.93** | **1.08** | 0.84 | **0.95** | **37.1** | **0.465** | **0.2832** | **0.5317** | 11.37 |

(21.8%). Nevertheless, our method still attained a slightly higher win rate of 61.6% compared to MMAudio-L-V2.

## 4.4 ABLATION STUDY

**Two-Stage Training Mechanism**  We employ a two-stage training strategy to optimize model performance (Table 4). For each 10-second video-audio clip, the first 8 seconds of audio are consistently used as the training target. In Stage 1 (Figure 3a), the random sampled 2-second segment of the target audio serves as the acoustic condition, achieving $FD_{PaSST}$ of 80.07 – this indicates the model might simply "copy-paste" conditional audio. In Stage 2 (Figure 3b), switching to the non-overlapping final 2-second audio as the condition significantly reduces $FD_{PaSST}$ to 56.00 (↓30.1%) and optimizes $KL_{PANNs}$ from 1.03 to 0.95, demonstrating that the model learns to leverage inherent self-similarity characteristics of video clips rather than mechanical replication.

**Subset Finetuning Strategy**  By finetuning on a high-quality audiovisual subset of VG-GSound (Chen et al., 2020) (selected via ImageBind score >0.3) for 40k iterations, the model achieves optimal semantic consistency (IB↑37.1) and temporal synchronization (DeSync↓0.465, Onset Acc.↑0.2832 and Onset AP↑0.5317) (Table 4). Compared to the non-finetuned version, spectral distortion (MCD) remains stable at 11.37, indicating that this strategy effectively enhances cross-modal alignment while preserving audio quality.

**Average Pooling**  Considering that taking the average pooling for conditional audio may remove some acoustic features, we compare the performance of our average-pooling and attention-based pooling. Table 5 shows that the two methods yield comparable results. We choose average pooling as it provides better training stability and lower computational cost. Additionally, experiments show that important acoustic features such as timbre, pitch, and rhythmic patterns can be well preserved after average pooling.

Table 6: Results when we mask out different conditioning components during inference.

| Method | Distribution matching | | | | | Semantic | Temporal | | | Spectral |
|---|---|---|---|---|---|---|---|---|---|---|
| | $FD_{PaSST}\downarrow$ | $FD_{PANNs}\downarrow$ | $FD_{VGG}\downarrow$ | $KL_{PaSST}\downarrow$ | $KL_{PANNs}\downarrow$ | IB↑ | DeSync↓ | Onset Acc.↑ | Onset AP↑ | MCD↓ |
| w/o. audio | 64.90 | 8.59 | 3.87 | 1.17 | 1.34 | 36.6 | **0.410** | 0.2619 | 0.5095 | 14.59 |
| w/o. sync | 90.63 | 6.96 | 1.17 | 1.12 | 1.19 | 32.5 | 1.240 | 0.2100 | 0.4925 | 11.71 |
| w/o. video | 55.86 | 4.90 | 1.13 | 0.85 | 0.96 | 36.9 | 0.471 | 0.2589 | 0.5117 | 11.36 |
| w/o. text | **55.63** | **4.87** | 1.11 | 0.85 | 0.96 | 36.8 | 0.474 | 0.2576 | 0.5123 | **11.36** |
| Ours | 56.00 | 4.93 | **1.08** | **0.84** | **0.95** | **37.1** | 0.465 | **0.2832** | **0.5317** | 11.37 |

**Multimodal Conditioning Components**  In our multimodal conditioning mechanism, each modality plays a complementary role. Text and video provide stable, high-level semantic, audio provides acoustic cues, and the sync features preserve frame-level alignment. This design allows the model to maintain global controllability (consistent timbre/semantic intent) and fine-grained temporally alignment. Table 6 shows that multi-modal information is complementary and necessary. Discarding any modality would result in significant losses in specific task dimensions (especially when removing audio or sync), while our approach achieves optimal overall performance.

## 5  Conclusion

We present AC-Foley, a novel audio-conditioned framework for video-to-audio generation that enables precise acoustic control through direct audio conditioning. By leveraging a two-stage training strategy, our approach effectively addresses critical challenges such as temporal adaptation and acoustic fidelity preservation, allowing reference sounds to be intelligently transformed and aligned with visual contexts. Extensive experiments demonstrate notable improvements over both text-conditioned baselines and video-conditioned methods, achieving superior control precision and audio quality. These advancements pave the way for new possibilities in creative sound design, particularly for applications requiring fine-grained acoustic variations that closely match visual events.

## Ethic Statement

Our experiments include a human study, which was conducted solely as an online user study. All participants participated voluntarily, and after obtaining informed consent. We note that malicious actors could potentially combine our system with video generation models to create synchronized audiovisual forgeries. To mitigate this risk, we will implement a safeguard by releasing our model under the Apache 2.0 license with explicit ethical use prohibitions when we are ready.

## Acknowledgement

The work described in this paper was partially supported by a grant from the Research Grants Council of the Hong Kong Special Administrative Region, China (Project Reference Number: AoE/E-601/24-N).

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

## A  TRAINING DETAILS

We train our model using the AdamW optimizer (Kingma & Ba, 2014; Loshchilov & Hutter, 2017) with an initial learning rate of $10^{-4}$, implementing a linear warm-up schedule for the first 1K steps across 260K total iterations at a batch size of 320. The learning rate undergoes scheduled decay: first to $10^{-5}$ after 200K iterations, then to $10^{-6}$ after 240K iterations. For model stabilization, we employ post-hoc exponential moving averaging (EMA) (Karras et al., 2024) with a consistent relative width parameter $\sigma_{\mathrm{rel}} = 0.05$ across all models. To optimize training efficiency, we utilize `bfloat16` mixed-precision computation and precompute all audio latent representations and visual embeddings offline for efficient loading during the training process. The training was conducted on 8 NVIDIA H800 GPUs and completed in roughly 26 hours.

## B  NETWORK DETAILS

Our model generates 44.1kHz audio encoded as 40-dimensional, 43.07fps latents. The transformer employs an architecture with 7 multimodal blocks followed by 14 single-modal blocks and a hidden dimension of 896.

## C  HUMAN STUDIES

**Videos and Reference Audios**    We manually selected 16 high-quality videos from the VGGSound test set (Chen et al., 2020), which cover a variety of categories and contain clear, easily perceivable temporal actions. For each video, we used the last 2 seconds of audio from the original 10-second clip as the conditional reference audio, with the corresponding category name serving as the text prompt to generate the audio for the first 8 seconds of the original video.

**User study survey.**    In the survey, participants watched and listened to 16 pairs of videos with generated audio, each with a real video for reference, comparing our method with MMAudio-L-V2 (Cheng et al., 2024). We performed a single-choice experiment where we randomized the presentation order of the video pairs. For each video pair, participants were asked to respond to two questions: 1) Please select the video below whose audio is most similar to this video (real video). 2) Which of the above options (two videos with generated audio) has the best audio sync with the video? The first question evaluates the acoustic fidelity between the generated audio and the ground truth audio. The second question evaluates the temporal alignment between the audio and video. We show a screenshot of our user study survey in Figure 5.

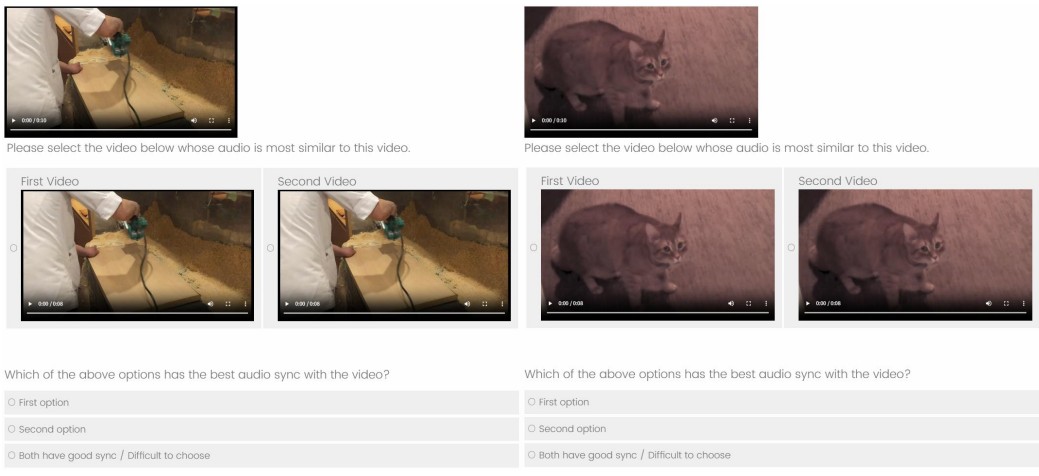

Figure 5: Screenshot of user study survey.

Table 7: Comparison of Mel-Cepstral Distortion for Foley generation using different conditional audio versus without conditional audio.

| Method | Mel Cepstral Distortion (MCD)↓ | | | | |
|---|---|---|---|---|---|
| | Ref. A | Ref. B | Ref. C | Ref. D | Ref. E |
| Without audio | 20.95 | 16.12 | 15.56 | 22.74 | 15.83 |
| With audio | **18.24** | **11.96** | **14.43** | **12.20** | **10.85** |

## D  MORE ABLATION STUDY

**Reference Audio Control**  To validate the effectiveness of our conditional audio mechanism, we conduct a controlled experiment on the VGGSound test set (Chen et al., 2020). Five distinct audio clips are randomly selected from the WavCaps dataset (Mei et al., 2024a), each truncated to the first 2 seconds as universal conditional references. For every test video, we generate five audio samples conditioned on these five references. We compute the Mel Cepstral Distortion (MCD) between each generated audio and its corresponding conditional reference to measure the acoustic (Table 7). As a baseline, we replace the conditional audio with a learnable null embedding vector (initialized as zeros and optimized during training) while retaining the same video inputs, then generate audio samples and calculate their MCD against the original 5 reference audios. This design isolates the impact of conditional guidance by comparing identical video inputs with and without referential control under fixed acoustic targets.

## E  LIMITATIONS

While AC-Foley demonstrates strong performance in single-source sound control scenarios, our method exhibits limitations when handling complex auditory environments. When input videos and conditional audio contain multiple concurrent sound sources (e.g., overlapping dialogue, ambient noise, and object interactions), the model may struggle to align specific sound elements with their corresponding visual triggers precisely. Additionally, extreme temporal mismatches between reference sounds and visual content (e.g., conditioning slow cat meowing sounds on video showing rapid keyboard typing) may lead to suboptimal generation quality due to conflicting rhythmic patterns.

## F  DATASET LICENSES

The following datasets were used in this work, along with their corresponding licenses:

1. VGGSound (Chen et al., 2020): Creative Commons Attribution 4.0 International (CC-BY 4.0).

2. AudioCaps2.0 (Kim et al.): MIT license.

3. WavCaps (Mei et al., 2024a): Creative Commons Attribution 4.0 International (CC-BY 4.0).

## G  LLM USAGE

During the writing process, the authors used a large language model (LLM) solely for language polishing and grammatical/style improvements. The LLM did not contribute to research ideation, experimental design, data collection, analysis, or the substantive academic content of the paper. The authors take full responsibility for the final text and for all claims made in the manuscript. The LLM is not listed as an author.

