# OpenReview forum: "AC-Foley: Reference-Audio-Guided Video-to-Audio Synthesis with Acoustic Transfer"
_ICLR.cc/2026/Conference — ICLR 2026 Poster_

### Official Review · Reviewer_jtQU · 2025-10-26

**Soundness:** 3
**Presentation:** 3
**Contribution:** 3
**Rating:** 6
**Confidence:** 3

**Summary:**

The authors propose AC-Foley, which directly addresses the expressive limitations that arise when using only text or audio semantic information, enabling fine-grained audio control. During training, the model adopts both approaches: conditioning on parts of the target audio and conditioning on segments that do not overlap with the target audio. By structuring this in a two-stage process, the model is explicitly trained to learn both aspects. This approach ensures that the generated audio is well-synchronized while avoiding overfitting to the conditioned audio, allowing for better generalization. Trained in this manner, the resulting multimodal transformer, AC-Foley, can generate long audio sequences of up to 8 seconds and demonstrates superior performance compared to other video-to-audio models.

**Strengths:**

1. The experiments are diverse. They convincingly demonstrate the validity of AC-Foley through a variety of baseline methods and a wide range of metrics, and the ablation study shows that the methodology in the paper is rigorously designed.

2. The authors clearly pointed out the limitations of existing audio synthesis methods that rely solely on text or semantic features, and their idea of directly conditioning on acoustic features is particularly commendable.

3. The two-stage curation in training to ensure that the model effectively learns the newly introduced method of “direct audio conditioning” is meaningful. Based on the experimental results, it appears to be an efficient training strategy that allows the multimodal transformer to adequately understand this new form of conditioning. With the authors’ approach, the model becomes both generalizable and aligned with the conditioning audio.

**Weaknesses:**

1. It is not sufficiently explained whether using overlapping or non-overlapping audio from the same video during training. It seems unlikely that the model would learn the ability to appropriately modify the conditioned audio to match the video. The paper does not clearly explain the basis for generating semantically aligned audio when conditioned on audio from a different video. In particular, for the timbre transfer shown in Figure 1, while the output is temporally aligned, it is difficult to judge whether it is semantically aligned.

2. Reduced complexity of generated audio is another limitation of this work. Since the model is trained using conditioned audio, as the authors themselves note in the limitations, the complexity of the audio the model can generate is likely constrained and cannot easily exceed that of the conditioned audio.

**Questions:**

Please explain how the model can generate semantically well-aligned audio even when conditioned on a random audio input. (Related to Weaknesses No.1)

---

> ### Author Response · Authors · 2025-11-20
>
> Thank you for your thoughtful review and constructive feedback. Below are our responses to your questions and concerns:
>
> **W1. Use overlapping or non‑overlapping training; how semantic alignment is learned when conditioned on audio from a different video**
>
> As mentioned in **Section 3.4**, we use a **two‑stage curriculum**. Stage I trains with **overlapping reference segments** (easier) so the model learns to reproduce and encode timbre and fine acoustic detail; Stage II trains with **non‑overlapping reference** segments (harder) so the model must transfer that learned timbre onto unseen temporal positions and rely on visual cues for event timing and semantics. This explicit split prevents the model from simply copying the input audio and enforces temporal adaptation.
>
> Regarding how semantic alignment arises despite a “random” reference, we believe this point stems from a misunderstanding of the experiment’s objective. When the reference audio comes from a different clip (i.e., its semantic content does not match the silent input video), the task is a **timbre‑transfer task**. Our goal is to generate an audio track that **temporally aligns with the input video** while **matching the timbre of the generated sound as closely as possible to the provided reference**, rather than having the **generated audio semantically align with the video**.
>
> For a concrete example, please see the **last video in the supplementary material and our reply to reviewer k8xt’s weakness 3**, which demonstrates this intended interpretation and shows how our output is judged as a timbre‑transfer success.
>
> **W2. Reduced complexity of generated audio because of conditioning**
>
> With conditional audio, we can achieve **precise, interpretable control over timbre and micro‑acoustic properties**, which is harder to achieve with text or high‑level semantic encoders alone. It also enables **creative workflows to create novel Foley effects**.
>
> To improve the robustness for more complex generation, we are exploring: (i)**multi‑instance conditioning**, where multiple reference latents are associated with detected visual objects or tracked regions (detected by an object detection model like yolo); (ii)**combining text and multiply audio conditions** to achieve more complex and fine-grained control.

---

### Official Review · Reviewer_nvKW · 2025-10-29

**Soundness:** 2
**Presentation:** 2
**Contribution:** 2
**Rating:** 4
**Confidence:** 4

**Summary:**

The paper introduces AC-Foley, a framework for video-to-audio synthesis. The model introduces direct reference audio conditioning to bypass the semantic ambiguity of language and the lack of acoustic granularity in training datasets, aiming to provide fine-grained control over the characteristics of the synthesized sound. The framework is built upon a multimodal transformer and utilizes a two-stage training strategy to ensure that the generated sounds maintain acoustic fidelity to the reference while accurately adapting to the temporal structure and context of the video. When conditioning on reference audio, AC-Foley achieves state-of-the-art performance across metrics measuring distribution matching, semantic alignment, and spectral fidelity. Furthermore, the framework remains highly competitive with other advanced video-to-audio methods even when the audio condition is removed.

**Strengths:**

- The paper unlocks some interesting applications, such as fine-grained sound synthesis (e.g., varying footsteps based on surface material), timbre transfer (applying one sound's tone to a different visual event), and at the same time the framework remains competitive, matching or closely approaching existing state-of-the-art performance in standard video-to-audio tasks.
- The method supports variable-length audio conditioning and does not require the reference audio and generated audio to have identical durations, unlike prior approaches.
- Overall, the paper is written well and easy to follow.

**Weaknesses:**

- Overall the task itself is not too challenging and therefore the technical contribution is relatively limited. Most components are reused from previous works, and adding additional audio control seems straightforward with the help of all state-of-the-art modules. The method relies on a multimodal conditioning vector that integrates information from video, text, and audio. This vector modulates the transformer input using adaLN layers, which is a common conditional technique and utilized in other generative models. The novelty lies in what is fed into the conditioning vector (i.e., acoustic features via VAE) rather than the conditioning mechanism itself.
- As the authors mentioned, the proposed method may have trouble when dealing with complex sound environments. When the input videos and conditional audio contain multiple concurrent sound sources (such as overlapping dialogue, ambient noise, and object interactions), AC-Foley may struggle to precisely align specific sound elements with their corresponding visual triggers. Moreover, it does not seem straightforward to me that this challenge can be easily addressed by the proposed framework.

**Questions:**

- What architectural or training modifications are being considered to improve the model's ability to isolate and precisely align specific sound elements to their corresponding visual triggers in complex auditory environments?
- Can the authors detail which specific acoustic features (beyond general timbre, such as transient impacts, decay characteristics, or harmonic content) are most successfully encoded by the VAE latents, enabling the fine-grained control that text and semantic encoders like CLAP fail to capture?
- The investigation into the ground truth DeSync mismatch (0.558s) suggested that both AC-Foley and baselines might be over-optimizing for the Synchformer metric, and that the metric's 4.8-second context window is inadequate for the 8-second video segments used. Have the authors explored alternative or complementary synchronization metrics that can accurately capture long-term temporal alignment across the full 8-second duration of the generated output?

---

> ### Author Response · Authors · 2025-11-20
>
> Thank you for your thoughtful review and constructive feedback. Below are our responses to your questions and concerns:
>
> **W1. Limited novelty/reuse of components**
>
> Our most significant contribution is our **novel task formulation and demonstrated effect**: we introduce a novel, practically useful conditioned video→audio (V2A) task that **enables precise, variable‑length reference-audio control and timbre transfer for silent video**. This task formulation and the applications it enables have been **noted positively by multiple reviewers (8Hbv, k8xt)**. Moreover, our method achieves the **state-of-the-art results**, especially in audio quality (**20% lower Fréchet Distance** and **28% lower Kullback–Leibler distance**) and acoustic fidelity (**22% lower Mel Cepstral Distortion**).
>
> Our technical novelty lies in enabling **precise, variable‑length audio conditioning** by average pooling and dynamic sequence length, and in the **two‑stage training** that forces temporal adaptation and timbre transfer. The **VAE‑based acoustic latents** we feed into the conditioning vector capture micro‑acoustic detail that semantic/text encoders cannot. The **two‑stage curriculum** (overlap → non‑overlap) is also a core methodological contribution that improves alignment and fidelity.
>
> We believe this task and the demonstrated improvements will meaningfully inspire follow‑on research and creative applications that demand controllable, temporally adaptive audio synthesis.
>
> **W2\&Q1. Difficulty in complex auditory environments**
>
> This task (robust source separation and precise per‑source alignment in highly concurrent, noisy scenes) is not the main focus of this paper, so the decrease in robustness in such settings **does not undermine our core contribution**. AC‑Foley’s primary goal is to demonstrate that **variable‑length reference audio conditioning, together with our two‑stage training**, reliably provides **fine‑grained timbre control and temporal adaptation** in typical single‑source or dominant‑source scenarios, which we validate across multiple datasets and metrics.
>
> To improve isolation and alignment in highly concurrent scenes, we are exploring: (i)**multi‑instance conditioning**, where multiple reference latents are associated with detected visual objects or tracked regions (detected by an object detection model like yolo); (ii)**MoEs**, which independently generate sounds of different types/sources; (iii)training with data of **increasing scene complexity**.
>
> **Q2. Which acoustic features are encoded by the VAE latents**
>
> The VAE latents can encode most acoustic features that influence perceived timbre and microacoustics, including **pitch, loudness, transient impacts, decay characteristics, and harmonic content**. In contrast, CLAP/semantic embeddings, which abstract to **class/semantic concepts**, lose all the detailed information mentioned above. This is because the training objectives of VAE and CLAP/semantic encoders are distinct. The VAE is trained with a **reconstruction loss**, while CLAP is trained with **semantic matching**. Therefore, VAE enables the fine-grained control that text and semantic encoders like CLAP fail to capture.
>
> **Q3. Alternatives to DeSync for long‑term alignment**
>
> Thanks for your suggestion. We added Onset-based evaluations for temporal alignment measures as follows:
>
> | Method | $\text{Onset Acc.}$ ↑ | $\text{Onset AP}$ ↑ |
> | :---- | :---- | :---- |
> | **Video-Foley** | 0.2146 | 0.3409 |
> | **MMAudio+Clap** | 0.2511 | 0.5107 |
> | **AC-Foley (Ours)** | **0.2832** | **0.5317** |
> | **V-AURA** | 0.2188 | 0.4880 |
> | **Foley-Crafter** | 0.2033 | 0.5312 |
> | **Frieren** | 0.2239 | 0.4689 |
> | **MultiFoley** | 0.2431 | 0.5173 |
> | **ThinkSound (w/o. CoT)** | 0.2735 | 0.5189 |
> | **HunyuanVideo-Foley** | 0.2671 | 0.5271 |
> | **MMAudio-L-V2** | 0.2816 | 0.5257 |
> | **AC-Foley (w/o. audio)** | 0.2619 | 0.5095 |

---

### Official Review · Reviewer_k8xt · 2025-10-31

**Soundness:** 4
**Presentation:** 3
**Contribution:** 4
**Rating:** 6
**Confidence:** 4

**Summary:**

This paper proposed AC-Foley, a novel conditioned Foley audio generation dataset that supports both text and audio conditioning. The proposed method uses a two-stage training paradigm to force the flow matching model to learn the multi-modal condition fused with text, video, and conditional audio. The proposed method surpassed existing SoTA on multiple datasets under different settings. In the Human evaluation, AC-Foley shows an over 70% winning rate against the SoTA baseline.

**Strengths:**

1. The idea of two-stage training is very interesting and has proven to be effective. AC-Foley separates the learning of audio features and audio-visual alignment into two stages. Starting from an easier setting with conditions from the same clip, and then extending to a harder case with unseen conditioning. This paradigm is quite inspiring.

2. This paper provides a rigorous evaluation against multiple baselines and different datasets. It not only outperforms in the audio-conditioned case, but also performs well on text-to-audio generation.

3. The provided example video includes some very nice results; the quality of the generated audio is very impressive.

**Weaknesses:**

1. The proposed method uses a multi-modal condition embedding in the flow-matching transformer. However, this multi-modal condition is fused from 3 different modalities (and one more time embedding), which naturally raises a concern about the quality of this conditional embedding. While the final results show that the model works well, the reviewer still wants to know what the intuition is behind this design, and what if we just use part of the conditional signals?

2. While the human study shows a much better preference against MMAudio-L-V2 in terms of acoustic fidelity, this human study only uses 16 selected results from VGGSound. This is a very small evaluation set, and it may lead to some bias and weaken these results.

3. During the evaluation, the paper only uses the last 2 seconds of the 10-second-long video, which could also introduce some bias. For example, the last few seconds may contain fewer actions than the previous ones, leading to a simplified task setting.

4. While the quantitative results look fine overall, the last example on the Greatest Hits is not very good. The proposed method generates a scratching sound while the action in the video is actually hitting. This may indicate that the model may fail when the conditioning sample is dramatically different from its silent input video.

**Questions:**

1. The paper uses Synchformer to measure the quality of audio-action alignment, which may not be the best choice (as mentioned in the paper, the Synchformer is biased by itself). Is it possible to use some other metrics to measure this alignment? For example, the onset comparison used in CondFoley, where one just needs to use the traditional onset detection method and compare against the GT audio.

2. It would be great if there could be some more examples of failure cases, as this can help better understand the limitations of the proposed method.

---

> ### Author Response · Authors · 2025-11-20
>
> Thank you for your thoughtful review and constructive feedback. Below are our responses to your questions and concerns:
>
> **W1.  Quality and intuition of the fused multi-modal conditional embedding**
>
> The fusion design is intentional: each modality plays a complementary role. Text and video provide **stable, high-level semantics**, audio provides **acoustic cues**, and the sync features preserve **frame-level alignment**. We want to fully leverage the distinct information from the three modalities to improve Foley synthesis. This design allows the model to **maintain global controllability (consistent timbre/semantic intent) and fine-grained temporal alignment**. For part of the conditional signals, we have made an ablation experiment as follows:
>
> | Method | $\text{FD}_{PaSST}$ ↓ | $\text{FD}_{PANNs}$ ↓ | $\text{FD}_{VGG}$ ↓ | $\text{KL}_{PaSST}$ ↓ | $\text{KL}_{PANNs}$ ↓ | $\text{IB}$ ↑ | $\text{DeSync}$ ↓ | $\text{MCD}$ ↓ | $\text{Onset Acc.}$ ↑ | $\text{Onset AP}$ ↑ |
> | :---- | :---- | :---- | :---- | :---- | :---- | :---- | :---- | :---- | :---- | :---- |
> | **w/o. video** | 55.86 | 4.90 | 1.13 | 0.85 | 0.96 | 36.9 | 0.471 | 11.36 | 0.2589 | 0.5117 |
> | **w/o. text** | **55.63** | **4.87** | 1.11 | 0.85 | 0.96 | 36.8 | 0.474 | **11.36** | 0.2576 | 0.5123 |
> | **w/o. audio** | 64.90  | 8.59  | 3.87  | 1.17  | 1.34  | 36.6  | **0.410**  | 14.59 | 0.2619 | 0.5095 |
> | **w/o. sync** | 90.63 | 6.96 | 1.17 | 1.12 | 1.19 | 32.5 | 1.240 | 11.71 | 0.2100 | 0.4925 |
> | **Ours** | 56.00 | 4.93 | **1.08** | **0.84** | **0.95** | **37.1** | 0.465 | 11.37 | **0.2832** | **0.5317** |
>
> These ablation results support our design intuition: multi-modal information is complementary and necessary. Discarding any modality would result in **significant losses in specific task dimensions** (especially when removing audio or sync), while our approach **achieves optimal overall performance**.
>
> **W2. Small evaluation set for human study**
>
> Thank you for pointing this out. We have increased the number of samples in the human study **from 16 to 32**, all from VGGSound. The evaluation protocol and questionnaire process remain consistent with the original experiment (each participant compared acoustic fidelity and time alignment in real audio and two generated audio samples). The expanded results are as follows:
>
> | Comparison | $\text{Temporal Win Rate}$ (%) | $\text{Temporal Tie Rate}$ (%) | $\text{Acoustic Win Rate}$ (%) |
> | :---- | :---- | :---- | :---- |
> | Ours vs MMAudio-L-V2 | **61.09**±4.29 | **21.77**±3.63 | **83.51**±3.35 |
>
> **W3. Using only the last 2 seconds of a 10-second video may bias the evaluation**
>
> Using the last 2s as the non‑overlapping reference **does not introduce a systematic bias toward easier generation**. In practice, 10s clips are commonly **extracted from longer continuous videos/audios**, so the final 2s are not inherently special relative to other segments of the same source, which means they **do not systematically differ in content or difficulty** because they occur at the end of the trimmed clip. Therefore, this selection does not create a bias.
>
> Finally, regarding the reviewer’s example, if the last 2s contain fewer actions, that does not simplify the task. In fact, it can make it harder because the reference carries less acoustic/event information while the model must generate an earlier 8s segment that contains more actions (i.e., more events to synthesize) from a sparser conditioning signal.
>
> **W4. Failure case on “Greatest Hits” (scratching vs hitting), indicating possible failure when conditioning is very different from the video**
>
> Thank you for pointing this out. However, we think there may have been a slight misunderstanding about the experiment's purpose. This example was for testing timbre transfer. The goal is to generate an audio that **temporally aligns with the input video** (drumsticks hitting a wall) while **matching the timbre of the generated sound as closely as possible to the provided reference** (leaf scratching). In short, we want to **transfer the “scratching” sound to the "hitting" action** while maintaining the temporal alignment.
>
> The CondFoley output of the knocking sound indicates that it failed to follow the timbre reference (i.e., it still followed the original video's semantics), whereas our method generated a scratching sound that matched the reference timbre and was temporally aligned with the video action.

---

> > ### Author Response · Authors · 2025-11-20
> >
> > **Q1. Synchformer metric bias and alternative alignment metrics**
> >
> > We have added Onset-based evaluations for temporal alignment measures as follows:
> >
> > | Method | $\text{Onset Acc.}$ ↑ | $\text{Onset AP}$ ↑ |
> > | :---- | :---- | :---- |
> > | **Video-Foley** | 0.2146 | 0.3409 |
> > | **MMAudio+Clap** | 0.2511 | 0.5107 |
> > | **AC-Foley (Ours)** | **0.2832** | **0.5317** |
> > | **V-AURA** | 0.2188 | 0.4880 |
> > | **Foley-Crafter** | 0.2033 | 0.5312 |
> > | **Frieren** | 0.2239 | 0.4689 |
> > | **MultiFoley** | 0.2431 | 0.5173 |
> > | **ThinkSound (w/o. CoT)** | 0.2735 | 0.5189 |
> > | **HunyuanVideo-Foley** | 0.2671 | 0.5271 |
> > | **MMAudio-L-V2** | 0.2816 | 0.5257 |
> > | **AC-Foley (w/o. audio)** | 0.2619 | 0.5095 |
> >
> > **Q2. More examples of failure cases**
> >
> > We will add more failure examples to the project page after our work is open-sourced.

---

> > > ### Comment · Reviewer_k8xt · 2025-11-25
> > >
> > > I thank the reviewer for their detailed reply! All these new experiments are very helpful in resolving my concern. The new ablation experiment on the conditioning signal helps answer the contribution of information from each modality, and the new Onset prediction evaluation also demonstrated the advantage of the proposed method.
> > >
> > > Yet, my concern in weakness 1 is not only related to the necessity of each conditional signal, but more on the way that they were fused. As suggested by the authors, each modality of conditioning information serves a critical role in the final audio generation, and they control very different aspects of the final generation. I do not doubt this, but I am wondering if the proposed fusion method is really the best choice. Information fusion usually leads to some kind of information loss, especially considering that these conditioning signals carry very different information. So, what I really want to ask here is whether there will be a better way to make use of these signals, rather than simply fuse them into one single embedding?
> > >
> > > Of course, I understand the experiment shows the proposed method works well, and asking for a different fusion strategy is out of the scope of this work, so I am not asking for any additional experiments. I will keep my current positive score, and I think it is still a very nice work!

---

### Official Review · Reviewer_8Hbv · 2025-11-01

**Soundness:** 3
**Presentation:** 4
**Contribution:** 3
**Rating:** 8
**Confidence:** 2

**Summary:**

The paper proposes reference-audio–guided video-to-audio (V2A) generator that conditions generation on a short audio clip to produce synchronized Foley for a silent video. To achieve this, the authors use a shared conditioning vector built from (i) CLIP text/video embeddings, (ii) Synchformer sync features, (iii) and VAE latents of the reference audio. The model is trained in two stages: first stage uses an overlapping reference segment while uses a non-overlapping tail segment to force temporal adaptation.

**Strengths:**

- The problem setup is novel and interesting. use reference audio indeed provides more precise control over the generated audio.
- The qualitative results are solid and impressive and show well the advantage of the proposed method.
- The paper is well-written
- The method is simple

**Weaknesses:**

- The method is a bit less intuitive for me. The conditioning modalities use average pooling which collapses time but the temporal information seems to be well-preserved. Can the authors explain the reason for this? Have the author tried temporal representation for the temporal modalities (such as the video)
- Missing discussion with multiple related work on V2A (e.g Rhythmic Foley, SSV2A, AV-LINK)

**Questions:**

See above

---

> ### Author Response · Authors · 2025-11-20
>
> Thank you for your thoughtful review and constructive feedback. Below are our responses to your questions and concerns:
>
> **W1. Average pooling may collapse time**
>
> In our framework, average pooling is mainly used to extract **stable global/timbre-based conditions** (such as the semantic context of the reference timbre or video), while temporal alignment and fine-grained temporal information are handled by other **temporally-sensitive modules** (mainly the sync features). Therefore, global controllability is preserved without losing necessary temporal information. The main experiment results (Table 1) show that temporal information and synchronization are well preserved. So, we did not use temporal conditioning representations, but we will explore them in the future.
>
> **W2. Missing discussion with multiple related works**
>
> The **Rhythmic Foley** and **AV-LINK** codes are not currently open source.
> We tested the **SSV2A**, and the results are as follows:
>
> | Method | $\text{FD}_{PaSST}$ ↓ | $\text{FD}_{PANNs}$ ↓ | $\text{FD}_{VGG}$ ↓ | $\text{KL}_{PaSST}$ ↓ | $\text{KL}_{PANNs}$ ↓ | $\text{IB}$ ↑ | $\text{DeSync}$ ↓ | $\text{MCD}$ ↓ | $\text{Onset Acc.}$ ↑ | $\text{Onset AP}$ ↑ |
> | :---- | :---- | :---- | :---- | :---- | :---- | :---- | :---- | :---- | :---- | :---- |
> | **SSV2A** | 236.71 | 17.47 | 2.34 | 1.74 | 1.85 | 26.2 | 1.210 | 19.79 | 0.2116 | 0.3988 |
>
> Thank you for pointing this out. We will add discussions in our revised version.

---

> > ### Comment · Reviewer_8Hbv · 2025-11-27
> >
> > I appreciate the authors response.
> >
> > My concerns are addressed and I keep my recommendation for the paper acceptance.
> > I noticed that the authors didn't update the paper. Please update the paper with the provided results and discussion. This would help the reader understand more the work contribution.

---

> > > ### Author Response · Authors · 2025-11-28
> > >
> > > Thank you for your reminder. We have updated our paper to include the new results and discussion, and uploaded the revised version. We appreciate your helpful feedback.

---

### Meta-Review · Area_Chair_QnRA · 2025-12-13

**Summary:**

Three reviewers indicated the paper is above the acceptance threshold (scores 6,6,8) and one indicated it is below the threshold (score 4). The main concerns had to do with: (1) Lack of intuition/clarity regarding the conditioning mechanism and video-audio alignment (8Hbv, k8xt, jtQU); (2) Missing discussions/comparisons to some existing works (8Hbv); (3) Evaluation protocol (k8xt); (4) Lack of technical novelty (nvKW).

After the rebuttal, reviewers 8Hbv and k8xt indicated they maintain their positive view of the paper. No response was recorded from reviewers jtQU and nvKW.

Overall, the reviewers’ views are mostly positive and the AC agrees with the responses of the authors to the concerns brought up by reviewer nvKW, who was the only one with a score below 6.

**Reviewer Concerns:**

Most major concerns have been properly addressed and misunderstandings have been clarified by the rebuttal. Two concerns raised by nvKW regarding limited technical novelty and lack of illustration on more challenging scenarios, were not addressed in the paper. But the authors explained the novelty is in the task formulation itself and that more challenging scenarios are left for future work. The AC views the authors’ responses to these points as satisfying.

**Reviewer Scores:**

**8Hbv: score 8.**

This is also the original score. The reviewer stated the authors’ rebuttal addressed their concerns and they would like to keep the positive score.

**K8xt: score 6.**

This is also the original score. The reviewer indicated they would like to keep the positive score, despite some remaining lack of clarity regarding what is the best way to fuse multiple modalities.

**nvKW: score 6.**

The reviewer’s original score was 4, mainly because of two concerns: lack of technical novelty and difficulty in handling multiple sound sources. The rebuttal clarified that the novelty is in the task formulation itself and that handling multiple sound sources is not the main goal of the work. It is not clear whether those arguments would satisfy the reviewer, however it seems likely they would, at least partially. In any case, it seems the reviewer does not object that the paper is accepted.

**jtQU: score 6.**

This is also the original score. The reviewer’s main concerns were: (1) lack of clarity regarding the overlap with the reference segments during training and how this affects semantic alignment; (2) limited complexity of the generated audio. The rebuttal clarified that overlap is used in the first stage of training but not in the second, and that semantic alignment is not the goal of the method (but rather a sort of timbre transfer). Directions for how to handle more complex scenarios were mentioned but left for future work. Since the reviewer was positive in the first place, it seems likely this rebuttal would lead them to maintain the initial score.

---

### Decision · Program_Chairs · 2026-01-26

Accept (Poster)